# AI Chatbots in Digital Mental Health

Luke Balcombe

Australian Institute for Suicide Research and Prevention, School of Applied Psychology, Griffith University,
Messines Ridge Road, Mount Gravatt, QLD 4122, Australia; lukebalcombe@gmail.com

**Abstract:** Artificial intelligence (AI) chatbots have gained prominence since 2022. Powered by big data, natural language processing (NLP) and machine learning (ML) algorithms, they offer the potential to expand capabilities, improve productivity and provide guidance and support in various domains. Human–Artificial Intelligence (HAI) is proposed to help with the integration of human values, empathy and ethical considerations into AI in order to address the limitations of AI chatbots and enhance their effectiveness. Mental health is a critical global concern, with a substantial impact on individuals, communities and economies. Digital mental health solutions, leveraging AI and ML, have emerged to address the challenges of access, stigma and cost in mental health care. Despite their potential, ethical and legal implications surrounding these technologies remain uncertain. This narrative literature review explores the potential of AI chatbots to revolutionize digital mental health while emphasizing the need for ethical, responsible and trustworthy AI algorithms. The review is guided by three key research questions: the impact of AI chatbots on technology integration, the balance between benefits and harms, and the mitigation of bias and prejudice in AI applications. Methodologically, the review involves extensive database and search engine searches, utilizing keywords related to AI chatbots and digital mental health. Peer-reviewed journal articles and media sources were purposively selected to address the research questions, resulting in a comprehensive analysis of the current state of knowledge on this evolving topic. In conclusion, AI chatbots hold promise in transforming digital mental health but must navigate complex ethical and practical challenges. The integration of HAI principles, responsible regulation and scoping reviews are crucial to maximizing their benefits while minimizing potential risks. Collaborative approaches and modern educational solutions may enhance responsible use and mitigate biases in AI applications, ensuring a more inclusive and effective digital mental health landscape.

**Keywords:** human–artificial intelligence; AI chatbots; digital mental health; mental health care

## 1. Introduction

Artificial intelligence (AI) chatbots are intelligent conversational computer systems that think, learn and complete tasks in combination with humans or independently, using big data, natural language processing (NLP) and machine learning (ML) algorithms to expand their capabilities, improve productivity and provide conversation, guidance and support [1,2]. Also indicated as conversational agents or generative AI using large language models, they are a result of progress in the past 15 years in the fields of robotics, ML, AI models and NLP. AI chatbots became eminent since the launch of ChatGPT in November 2022 [3].

There are opportunities where AI chatbots can provide insightful responses beyond human capacity. However, they may lack a personalized and empathetic touch. It is proposed that human–artificial intelligence (HAI) may help overcome such limitations, whereby humans and AI enable each other's strengths to collaborate on a common task or goal for efficient, safer, sustainable and enjoyable work and lives. The HAI concept aligns with the Center of Humane Technology's work with integrating human values, such as empathy, compassion and responsibility in AI [4].

Mental health is a critical issue that affects many millions of people worldwide [5]. As an example, around 20% of Australian adults have a mental disorder, which increases to 44% when the experience of mental illness is considered over a lifetime [6], costing the economy billions of dollars [7] because of diminished health and reduced life expectancy [8]. Unfortunately, many people do not receive the help they need due to various barriers such as a lack of access to mental health services, stigma and cost [9–12]. Digital mental health solutions target young people with technology for mental health assessment, support, prevention and treatment [13]. For example, AI and ML models are used in the prediction of mental illness [14], and AI chatbots are used for psychological support [15]. However, there is uncertainty around the ethical and legal implications of these tools.

The aim of this narrative literature review is to demonstrate how the potential for AI chatbots to assist various populations with accessible digital mental health through relevant, scalable and sustainable data-driven insights is challenged by the need for creating ethical, responsible and trustworthy AI algorithms.

## 2. Methods

This narrative literature review is adapted from the four steps outlined by Demiris et al. [16]: (1) Conduct a search of numerous databases and search engines; (2) Identify and use pertinent keywords from relevant articles; (3) Review the abstracts and text of relevant articles and include those that address the research aim; and (4) Document results by summarizing and synthesizing the findings and integrating them into the review.

The heterogeneity of the topic prevented a systematic review. In addition, the topic is still evolving, and there were not enough studies that meet the strict criteria of a systematic review. Additionally, the field of digital mental health is interdisciplinary, incorporating aspects of psychology, technology and health care. This leads to a wide range of research approaches, methodologies and study designs, making it challenging to apply strict systematic review criteria. Instead, the purposively selected articles were presented in an educational approach to show how AI chatbots may impact digital mental health. By purposively selecting articles that aligned with the research aim and reviewing them in a comprehensive manner, valuable insights were gathered and are presented in a coherent narrative literature review. This approach allowed for flexibility in considering various perspectives and ideas within the topic, contributing to a more holistic understanding of the subject matter.

The selection of peer-reviewed journal articles, media articles and conference proceedings were retrieved from searches of computerized databases, purposive online searches and authoritative texts based on an assessment of three of the research questions posed in the Editorial, "AI Chatbots: Threat or Opportunity?" [3]. These research questions were used as a guide to explore the topic of interest, because it is not yet possible to arrive at a comprehensive understanding of the state of the science:

1.  The development of AI chatbots has been claimed to herald a new era, offering significant advances in the incorporation of technology into people's lives and interactions. Is this likely to be the case, and if so, where will these impacts be the most pervasive and effective?
2.  Is it possible to strike a balance regarding the impact of these technologies so that any potential harms are minimized while potential benefits are maximized and shared?
3.  A growing body of evidence shows that the design and implementation of many AI applications, i.e., algorithms, incorporate bias and prejudice. How can this be countered and corrected?

The database searches used were Scopus, ScienceDirect, Sage and the Association for Computing Machinery (ACM) Digital Library. The search engines used were PubMed, Google Scholar and IEEE Xplore. The search terms used were "AI chatbots" OR "generative artificial intelligence" OR "conversational agents" AND "digital mental health" OR "mental health care".

The following selection of criteria were used:

Inclusion criteria:

- Studies that have been published in peer-reviewed journals, media articles and conference proceedings.
- Studies that have been published in the English language.
- Studies that have been published between 2010 and 2023.
- Studies that have investigated the use of AI chatbots, generative artificial intelligence or conversational agents in digital mental health or mental health care.
- Studies that have reported on the effectiveness of AI chatbots, generative artificial intelligence or conversational agents in digital mental health or mental health care.

Exclusion criteria:

- Studies that are not published in peer-reviewed journals, media articles and conference proceedings.
- Studies that are not published in the English language.
- Studies that are published before 2010 or after 2023.
- Studies that do not investigate the use of AI chatbots, generative artificial intelligence or conversational agents in digital mental health or mental health care.
- Studies that do not report on the effectiveness of AI chatbots, generative artificial intelligence or conversational agents in digital mental health or mental health care.

Boolean operators such as AND and OR were used to combine search terms and refine search results. For example, using the Boolean operator OR between "AI chatbots" and "generative artificial intelligence" retrieved articles that contain either one of these terms. Similarly, using the Boolean operator AND between "conversational agents" and "digital mental health" retrieved articles that contain both these terms. Boolean operators helped to narrow down search results and make them more relevant to the research question.

Relevant articles and their reference lists were explored based on (1) relevance to the guiding research questions, (2) showing examples of theoretical and empirical research and development, and (3) highlighting issues and possible solutions. These articles were applied in a best-evidence synthesis for a complete, critical and objective analysis of the current knowledge on the topic. Overall, the method shows a systematic and transparent approach that minimizes bias by ensuring a comprehensive search, focusing on relevant articles and presenting a fair synthesis of findings. However, it is important to note that bias can still exist in the literature itself; therefore, the studies were critically evaluated, and any potential limitations or biases within the selected articles were acknowledged.

## 3. Results

### 3.1. The Impact of AI Chatbots on Technology Integration

Research Question 1: The development of AI chatbots has been claimed to herald a new era, offering significant advances in the incorporation of technology into people's lives and interactions. Is this likely to be the case, and if so, where will these impacts be the most pervasive and effective [3]?

The use of AI chatbots has the potential to bring significant advances and impact various aspects of people's lives and interactions [17], especially where human to human interaction is not preferred or possible to obtain [18]. AI chatbots may provide customer service and support, health care and mental health support, education and e-learning, personal productivity and assistance, language translation and communication as well as social companionship and entertainment [19,20]. The diversity of uses for AI chatbots and the large mix of empirical literature means it is reasonable to focus on one area.

Mental health care is a good example, because AI chatbots have been considered a viable resource in this domain for more than a decade [21]. There are promising clinical outcomes for AI chatbots providing relevant and continually accessible support [22,23] for depression in adults [24], anxiety in university students [25,26] and attention-deficit/hyperactivity symptoms for adults [27].

AI chatbots may help address the barriers to the help-seeking process for mental health issues by offering personalized, accessible, affordable and stigma-free assistance, promoting early intervention and generating valuable insights for research and policymaking [28–31]. AI chatbots may be particularly useful in monitoring, communication, memory assistance, screening and diagnosis, with the aim of understanding a patient's emotions and assisting in the analysis of large datasets. For example, algorithms may identify patterns and trends that might be missed by human analysts. By analysing a patient's medical history, genetic data and other relevant factors, algorithms could generate tailored symptom checks and treatment recommendations that consider the individual's unique needs and circumstances.

The opportunities for AI chatbots should also be considered in terms of the challenges posed, such as a lack of human connection, a reliance on technology, the accuracy and reliability of information, ethical and privacy considerations as well as misdiagnosis and limited understanding [28–31].

A 2023 overview of mental health chatbots found 10 apps on the market for a variety of mental health concerns (e.g., anxiety and depression) and users (e.g., rural dwellers, shift workers, students, veterans and adolescents), for a variety of aims (e.g., to improve social or job interviewing skills) [18]. The overview took interest in AI chatbots for their accessible, affordable and convenient social and psychological support. However, vulnerable users may overrate the benefits and encounter risks, especially during a crisis, because AI chatbots were allegedly incapable of identifying crisis situations. Therefore, poor semantics were found to undermine AI chatbots, because they were not developed enough to understand the context of users' words and failed to respond effectively or at all.

Users may not be aware of the difference between humans and humanlike chatbots. These limitations are human factors, of which education is the key for effectively collaborating to produce sustainable solutions [32]. Users and practitioners need guidance on the uses of AI chatbots, similar to what is generally required for digital mental health platforms and interventions [33].

The different fields of psychology, psychiatry, AI and health care, as well as educators, policymakers, computer scientists and technology developers working on mental health care, means there are significant challenges to overcome in order to realize overall benefits [34]. Mental health professionals and policymakers hold the key to AI chatbots being a useful tool in the intelligent system toolbox. However, it appears that graduate students and research scientists may best drive change through their willingness and ability to effectively collaborate with computer scientists and technology developers.

AI chatbots offer promise as complementary tools rather than a replacement for human mental health professionals [18,20]. A 2021 review of digital mental health interventions (DMHIs) found AI chatbots to speculatively help mental health professionals meet overwhelming service demand [34]. A 2023 systematic review and meta-analysis of randomized controlled trials (RCTs) found AI chatbots to be acceptable for a wide range of mental health problems [35]. For example, an RCT found a fully automated conversational agent, Woebot, to be a feasible, engaging and effective way to deliver cognitive behavioural therapy (CBT) for anxiety and depression in young adults [25]. There is promise for Woebot [36] and Wysa [37] in establishing a therapeutic bond with users.

Although AI chatbots are feasible as an engaging and acceptable way to deliver therapy, more studies are required for what may facilitate a digital therapeutic alliance [36,37] and to reduce misunderstandings [38]. Mental health chatbot attrition rates are lower in comparison to other digital interventions [24,39]. However, dropout rates require attention, as does clarity around what disorders they are useful for [40]. Some reviews found a high potential for AI chatbots in identifying patients at risk of suicide [41–43], and triage and treatment development through NLP integrated to social media in real-time [44–46].

Developments in Generative Pre-Trained Transformer (GPT) programs like ChatGPT 4 means AI chatbots may be used in suicide prevention [47]. However, there is a need for better understanding AI chatbot limitations such as negative sentiment, constrictive thinking, idioms, hallucinations and logical fallacies. A study of messages related to people's

suicidal thoughts sought insights from the arrangement of their words, the sentiment and rationale [48]. While AI chatbot hallucinations and fallacies require human intervention, it is possible to detect idioms, negative sentiment and constrictive language with off-the-shelf algorithms and publicly available data. However, safety concerns were publicized after a chatbot, Eliza, was blamed by a Belgian man's widow for her husband's suicide [49].

There is a need for qualitative studies to help reduce poor semantics and errors as well as increase trust in AI chatbots. For example, thematic analysis from retrospective data is required to identify common themes of messages sent to mental health chatbots in order to increase the effectiveness of AI chatbots as a source of support. AI chatbots may help improve problem areas through NLP for sentiment analysis, which is fast and effective qualitative data analysis, to assist in understanding multidimensional online feedback.

Recommendations for identifying and evaluating the impact of AI chatbots are as follows:

- Conduct qualitative studies using AI chatbots to demonstrate how they assist with accessibility, engagement and effectiveness through (1) identifying user needs, (2) understanding barriers to its use, (3) evaluating user experience and AI chatbot impact and (4) integrating human–AI approaches to overcome problem areas.
- Contribute to empirical evidence with longitudinal studies and RCTs to see which mental health conditions and populations AI chatbots may be recommended for.
- Determine a practical attrition prediction possibility to identify individuals at a high risk of dropping out through applying advanced machine learning models (e.g., deep neural networks) to the leveraging analyses of feature sets (e.g., baseline user characteristics, self-reported user context and AI chatbot feedback, passively detected user behaviour and the clinical functioning of users).

*3.2. The Balance between the Benefits and Harms of AI Chatbots*

This is difficult to answer on a global scale because of a lack of widely collaborative international standards in addition to the diversity of applications for AI chatbots. However, current investment in AI research, education, societal adaptation, innovation, employment opportunities and jobs creation appear to be insufficient upon considering the scale of the impending changes.

The novelty and complexity of AI in mental health means it is timely to focus on cutting-edge education such as specialized university courses in digital mental health and informatics that use peer-reviewed and routinely updated textbooks and modules. The intent should be to stimulate discerning skills and critical thought from a mix of subjects that will assist in pioneering benefits to mental health care and AI technology industries while also mitigating the increasing costs from mental illness. While AI chatbots are eminent, they have yet to reach their potential in assisting with mental health problems in digital users, who are predominantly young people [18].

Quality, effective and usable chatbots such as Woebot and Wysa are available to assist with mental health support [36,37]. However, various studies are needed to show evidence for a broader array of mental health disorders and symptoms. Furthermore, development is mostly being driven by the technology side with an interest in mental health rather than by mental health professionals who are technologically savvy. The differences in communication styles and methodologies between technology and mental health care researchers (i.e., pattern-based versus hypothesis-derived) has limited the combination of these approaches. Another hindrance is the limited opportunities for high-level researchers who are capable of understanding and implementing hybrid methods.

However, there is good potential for mental health care to serve as an example where AI chatbots may assist in providing (cost-)effective solutions for a range of users and aims [21–24,36–38]. Mental health care professionals may need to embrace AI chatbots for their use to become more productive [50]. There also needs to be conscious efforts to broaden the way in which productivity is measured if significant advances integrating technology into people's lives and interactions are to be realized. For example, how can AI

chatbots' contribution to the economy and the health of people be accurately measured? While there may be a gain to the gross domestic product (GDP) of developed countries, there may also be some job losses because of AI disruption. Although productivity, affordability and accessibility are important levers, so are policies that consider mental health and human capital.

The impact of AI chatbots on productivity needs to be considered in terms of national and international economics, standards and regulations. It is apparent that not all governments are aligned in principle. Also, the digital divide is questionable in terms of not further marginalizing the underserved and the unserved [34]. Therefore, productivity and humanity need to be considered in terms of global risks such as war and the costs that the physical effects of climate change will bring [50]. While some governments heavily invest in defence, the decarbonisation of heavy industries and transition among energy systems, there will be competing demands for investment into AI technologies. Meanwhile, the example of the emergence of ChatGPT shows the difficulties of stakeholders grappling with the pace of technological development.

It is unclear how the Productivity Commission's forecasts of AI boosting the Australian economy have been calculated to arrive at a predicted boon between a 66.67% and 266.67% higher GDP in the next decade [7]. In 2023, the Australian Government projected an outlook for the next 40 years in terms of intergenerational equity, with forecasts of higher financial burdens on younger generations [51]. This leads to the question of how such countries manage and maximize the major shifts that are underway in their national economy while also effectively integrating the impact of AI technologies.

In the example of mental health care in Australia, it is necessary to explore the existing structure for safety and quality to see AI's consistency with it before examining its economic potential. Australia's National Standards in mental health services provide a framework for safety and quality in hospitals and community services and are primarily intended to regulate the practice of mental health professionals [52]. Yet, with overwhelming demand and limited supply in mental health care exacerbated by further strain during the COVID-19 pandemic [53,54], digital mental health filled a service gap, culminating in the development of the National Safety and Quality Digital Mental Health Standards in 2020, which aimed to improve the safety and quality of digital mental health service provision [55]. However, mental health professionals and policymakers are currently confronting the opportunities and challenges of AI [56]. For example, prompt engineering is used with ChatGPT to bypass content filters in social media. This could lead to harm and problems such as exploiting vulnerability.

The Australian Government adopted a voluntary ethics framework in 2018 for "responsible" AI, in order to guide businesses and governments to responsibly design, develop and implement AI [57]. However, mainstream AI chatbots are mostly being developed in the US. Australia and the US are among the various countries seeking input or planning on AI chatbot regulation [58]. The EU implemented the Digital Services Act and the Digital Market Act, aiming to create a safer digital space where the fundamental rights of users are protected and to establish a level playing field for businesses [59]. There is a need to ensure that AI algorithms are developed and trained using diverse and representative datasets and that any insights generated by AI are rigorously validated and verified by human experts. ChatGPT's owners, OpenAI, suggested proactively managing the risks of these "frontier AI" models [60]. OpenAI initially proposed conducting pre-deployment risk assessments, external scrutiny of model behaviour, using risk assessments to inform deployment decisions and monitoring and responding to new information about model capabilities and uses post deployment.

It is essentially up to users to be transparent about their use of AI, take steps to protect privacy and confidentiality and take care to use it responsibly for optimising its performance [61]. For example, recruits may use AI chatbots to accomplish their duties, which detracts from the intent of seeking human input and raises critical questions about the value of cooperative work if trust cannot be established and maintained [62]. A

main problem with AI chatbots is that they are a new technology with the potential of becoming fundamentally pervasive in terms of cybersecurity risk because of their ability to create all sorts of malicious codes and algorithms that can cause infrastructure or financial system chaos [63].

The use of AI in mental health research has been well established as potentially yielding important insights and improving outcomes for individuals with mental health disorders [64,65]. However, it is important to carefully classify and regulate "high" risks and prioritise ethical considerations at every step. The increasing use of AI chatbots for mental health and crisis support means that stakeholders need to increase their attention and education in order to effectively leverage these tools [18,66]. For example, fair aware AI has been called for in digital mental health to promote diversity and inclusion [67], and explainable AI has been suggested as a tool for demonstrating transparency and trust between users and practitioners [68].

It is proposed that HAI may complement these concepts in an evolving AI system where multiple AI models work together with human input to generate recommendations and predictions rather than relying on a single algorithm. A next step is establishing optimal combinations of humans and AI chatbots for various tasks in research, practice and policy [69]. However, it is necessary to consider AI technologies overall in terms of plans for broad-ranging regulation.

According to Australia's AI Ethics Principles [70], regulation can help achieve safer, more reliable and fairer outcomes for all Australians; reduce the risk of negative impacts on those affected by AI applications; and encourage businesses and governments to practice the highest ethical standards when designing, developing and implementing AI. A subsequent position statement on generative AI suggested that regulation can help address concerns about potential harms such as algorithmic bias and errors, the spread of misinformation, inappropriate content and the creation of deepfakes [71].

By implementing measures such as transparency, accountability and risk mitigation strategies, regulation can help ensure that AI is used responsibly and ethically [72]. Furthermore, regulation can help raise public trust in AI technologies by ensuring that they are developed and used in a manner that is consistent with societal values and expectations [73]. This can help facilitate the adoption of AI technologies and enable society to fully realize their potential benefits.

The regulation of AI should include defining what constitutes "unsafe" AI and determining what aspects of AI should be subject to regulation [74]. This requires a clear understanding of the anticipated risks and benefits of AI technologies on a global scale as well as insights into the public's trust and the acceptance of AI systems. While people in Western countries are more cautious of AI and less confident that the benefits outweigh the risks, those in the emerging economies (i.e., Brazil, India, China and South Africa) are more trusting and embracing of AI, in addition to young, university-educated people as well as those in managerial roles [75].

Overly stringent regulations could stifle innovation and hinder the development of AI technologies [76]. As such, regulation requires international cooperation to be truly effective. Without a global consensus, companies might simply migrate their AI development activities to less regulated jurisdictions, leading to a regulatory race to the bottom. There is a need to secure AI models and their associated systems by using industry-standard security protocols. AI models and systems should be regularly updated and patched to address any discovered vulnerabilities.

Recommendations for regulating and/or promoting the responsible use of AI applications are as follows:

- Invest in research to evaluate the efficacy and potential harms of AI applications and develop systems to monitor and audit AI systems for unusual or suspicious activity.
- Implement rigorous safety measures, robust regulations and collaborative standards to ensure the responsible use of AI technologies.

- Validate a HAI model combining AI chatbots with human experts in research, practice and policy to optimise mental health care assistance.

### 3.3. The Mitigation of Bias and Prejudice in AI Applications

The World Health Organisation offered a warning, stating that the use of generative AI for health care must be approached with caution [77]. AI algorithms are only as good as the data they are trained on, and biases in the data can lead to biased results [78]. Additionally, the use of AI in mental health care raises important risks and ethical considerations [79] as well as security, bias and privacy concerns, particularly when it comes to the storage and use of sensitive medical and personal data [80].

More generally, there are "high-risk" uses of AI and automated decision-making, which warrant the warning of potential harms, including the creation of deepfakes and algorithmic bias [81]. There is also concern about AI perpetuating or amplifying biases or narrow viewpoints [82,83] as well as automating jobs and therefore replacing humans in some capacities [84]. However, AI can be used to counter disinformation and to improve the accuracy and reliability of reporting [85,86]. The challenge lies in defining and deciding what constitutes "unsafe" AI. Various Australian science experts have called for rigorous safety measures, robust regulations and standards to be implemented for these "unsafe" AI [76]. It is apparent that mitigating measures for high-risk AI should be quickly and proactively sought to avoid hampering progress in AI.

Generative AI is being used in the media to create more personalised and targeted advertising, to automate content creation and curation and to analyse audience behaviour and preferences [87,88]. Misinformation or disinformation may stem from tools like Chat-GPT [87] in combination with social media, which produce the mass de-prioritization of legitimate news outlets in favour of spam as well as false or manipulative user-uploaded content [87]. Bias and errors in generative AI [67,87] highlight the questionability of existing information assessment guidelines because of evidence credibility, source transparency and limitation acknowledgment. Generative AI has shown the need for new guidelines to promote ethics, fairness, privacy and transparency [76] as well as recognize the intellectual property rights of human creators and organizations [89]. This may be exacerbated by potentially anticompetitive practices used by dominant technology platforms such as Google and Meta [88].

There is a need to counter and correct the AI applications that help perpetuate bias, harassment and marginalization as well as the loss of critical thinking and independent thought. AI chatbots may be a part of innovative solutions to answer calls for the detection and moderation of fake news [90] and the transparent regulation of social media platforms [91–93]. As an example, a review of the impact of YouTube on loneliness and mental health found its recommendation algorithms may inadvertently reinforce existing beliefs and biases, spread misinformation and disinformation as well as enable unhelpful or harmful content [46]. However, the review also found that YouTube can have positive effects on loneliness, anxiety and depression if users actively engage with the platform and use it as a tool for education, social connection and emotional support.

There are opportunities for biased and prejudiced AI applications to be countered and corrected through education and research with the assistance of AI chatbots [94]. However, human researchers/experts who understand the history and context of research problems may need to help prompt and supervise AI chatbots for solutions. For example, YouTube's recommendation algorithm is designed to suggest videos based on users' viewing history, search queries and other data points [95]. Since YouTube's launch in 2005 up until 2011, it was designed to recommend videos that attracted the most views or clicks. Then, in 2012, it was designed to directly respond to metrics such as shares, likes and, to a lesser extent, dislikes. From 2016 onwards, it was designed to increase safety, with efforts made to remove dangerous content and demonetise those who did not follow regulations. However, the development of AI chatbots means that continuous adaptation is critical through legislation and setting ethical values [94] in addition to improving current AI systems [46].

YouTube has initiated mental health policies, algorithm changes, content moderation, content creator and user psychoeducation, mental health and crisis resource panels, self-harm and suicide content warnings and parental controls and settings [96]. YouTube reported largely reduced borderline content as it immediately removes offensive content where detected [97]. However, the algorithm can also create filter bubbles and echo chambers where users are exposed to content that reinforces their existing beliefs and biases [98]. This can lead to polarisation and misinformation, which can have negative consequences for mental health. Improved algorithms are called for to detect bias and errors as well as moderate how videos appear in a watch list in order to steer users to safe, well-informed and inclusive content as well as refer them to mental health and crisis resource panels with suitable information and resources, in conjunction with assistance from AI chatbots [46].

However, problematic social media use in young people affects one in three individuals in the Australian youth, and it is not only limited to YouTube [99]. For example, cyberbullying is also an issue in other social media (e.g., Facebook, Twitter, Snapchat, Instagram, TikTok) [100]. Various studies have found a clear link between heavy social media use and an increased risk for depression, anxiety, loneliness, self-harm and suicidal thoughts [101–103]. Although there is a lack of psychological studies on TikTok [104], a causal study across American colleges found that access to Facebook led to an increase in severe depression by 7% and anxiety disorder by 20% [102]. This significant link between the presence of Facebook and a decline in mental health in young people is concerning when considering the fact that the arrival of Facebook in 2004 was followed by an increase of 57% in deaths by suicide among Americans aged 10–24 between 2007 and 2017 [105].

Major data breaches and the use of "psychological warfare tools" on Facebook were reported in 2018 with the Cambridge Analytica files [106]. After calls were sounded for data to be used following principles of ethics, privacy and security, Australia took the international lead in social media regulation with the Online Safety Act 2021 after public hearings revealed that Facebook's algorithms were potentially harmful and unsafe [107]. However, in 2022, the Australian government and the technology industry realized that an outdated classification system hindered the creation of new codes for regulating online content [108]. In 2023, Australia expressed interest in pursuing risk-based classification systems for AI chatbots as were being drafted in Canada and the EU [109].

Advances in AI chatbots and other tools such as predictive models and virtual assistants means that multiple models may be combined with human expert input to address mental health challenges and suicide prevention, improve access to care and reduce the barriers to seeking help. These tools use NLP and ML to mine mental health data, to understand and respond to individuals' needs and to provide personalised support. A theoretical framework proposed an adaptive Social Media Virtual Companion (SMVC) for educating and supporting adolescent students in interactions in social media environments in order to achieve a measure of collective well-being [110]. This SMVC framework is an example of how to design social media systems and embedded educational interventions through HAI, because automatic processing powered by a recommendation algorithm is combined with educator/expert intervention and guidance.

HAI mental health strategies are proposed to be useful for the design and development of a multi-model responsible social media system in education settings. For example, an adaptive SMVC may assist in promoting the obtaining of more balanced and diverse content as well as reducing the impact of bias and errors in algorithmic recommendation systems such as filter bubbles and echo chambers. By integrating off-the-shelf solutions like Viable for sentiment analysis and DataMinr for monitoring and analysing social media, the SMVC system can learn from HAI feedback and recent data to adjust recommendations accordingly.

However, AI-generated sentiment affects the emotional language used in human conversation, therefore potentially affecting social relationships. Randomized experiments found that algorithmic recommendation systems change how people interact with and

perceive one another socially; people are evaluated more negatively if they are suspected of using an algorithmic response from AI chatbots like ChatGPT [111]. Therefore, educators should proactively and transparently encourage the use of AI chatbots to avoid negative perceptions. Users may need to be taught how to be discerning and critical of the information AI chatbots provide and learn how to effectively leverage these tools to help solve complex problems in their studies as well as cautiously use them in self-care for mental health—obtaining assistance where required.

Recommendations for countering and correcting the flaws of AI applications are as follows:

- Vulnerable people need more informed guidance on how to self-manage their mental health when assisted by AI chatbots in order to connect with resources and treatments.
- Social media mental health and crisis resource panels may be enhanced by linking to AI chatbots that provide vetted digital mental health and crisis services or referrals as necessary.
- HAI mental health strategies with SVMC may be explored for cautiously navigating a safer, more responsible social media with humane, fair and explainable system recommendations.

## 4. Conclusions

This narrative literature review has explored the multifaceted impacts of AI chatbots on various aspects of society, particularly focusing on their potential in the field of mental health care. This review is useful for providing an overview of the topic, identifying gaps in the literature and generating new research questions. By synthesizing both theoretical and empirical research, this study provided a comprehensive overview of the current state of AI chatbots in mental health care. The evidence presented indicates that AI chatbots hold promise for revolutionizing mental health support, offering accessibility, engagement and effectiveness in assisting individuals and populations with a wide range of mental health concerns and aims. However, it is crucial to approach their implementation and regulation with caution and responsibility. The novelty of AI chatbots in mental health means that this narrative literature review shows examples of theoretical and empirical research that future studies can apply.

The development of AI chatbots brings opportunities for serving underserved and unserved populations, as well as blending care for the well-served, especially in treating common disorders such as anxiety and depression. However, there are challenges in knowing which AI chatbots are of good quality and which are useful and effective. Therefore, it is important for future research to clarify these areas as well as the level of care required for crisis support. The human factors of human–computer interaction require more attention through empirical research. AI chatbots offer accessible and convenient support, helping to address the barriers in the help-seeking process for mental health issues and have shown promise in various clinical trials. Nevertheless, limitations such as poor semantics, biases and the need for qualitative studies to improve user experience must be acknowledged and addressed. AI chatbots should be seen as complementary tools rather than replacements for human mental health professionals. Notwithstanding, there is a need for more empirical evidence and advocacy for users and practitioners to distinguish the quality, usability and effectiveness of AI as well as its uses and the populations that would benefit from them. If AI chatbots evolve to provide appropriate answers to these areas for clarification, then a more autonomous use of these tools will become progressively possible.

Furthermore, there is a need for regulation and responsible use of AI applications, given the potential for biases, privacy concerns and the amplification of misinformation. This emphasizes the importance of international collaboration in establishing standards and regulations to ensure the ethical and transparent use of AI technologies. The balance between innovation and regulation must be carefully struck in order to avoid stifling progress while safeguarding against potential harm.

Additionally, the review highlights the role of collaborative AI in countering biases and errors in AI applications, especially in the context of social media and mental health support. By integrating human expertise and sentiment analysis into AI models, it becomes possible to provide more balanced and diverse content while reducing the impact of algorithmic biases.

Overall, the review shows promise for the use of AI chatbots in mental health care, but it also highlights the need for further research, such as scoping reviews, to evaluate their effectiveness and address the risk of bias and ethical concerns. It emphasizes the need for careful consideration, research and collaboration in harnessing the potential of AI chatbots. While they offer transformative possibilities in various domains, responsible development, regulation and ongoing evaluation are essential to maximize their benefits while minimizing risks. Collaborative efforts between technology developers, mental health professionals, policymakers, researchers and educators can help ensure that AI chatbots contribute positively to society's well-being and mental health support.

**Funding:** This research received no external funding.

**Institutional Review Board Statement:** Not applicable.

**Informed Consent Statement:** Not applicable.

**Data Availability Statement:** Not applicable.

**Conflicts of Interest:** The author declares no conflict of interest.

**Abbreviations**

| | |
|---|---|
| ACM | Association for Computing Machinery |
| AI | artificial intelligence |
| CBT | cognitive behavioural therapy |
| DMHIs | digital mental health interventions |
| EU | European Union |
| GDP | gross domestic product |
| GPT | Generative Pre-Trained Transformer |
| HAI | Human–artificial intelligence |
| HCI | human–computer interaction |
| IEEE | the Institute of Electrical and Electronics Engineers |
| ML | machine learning |
| NLP | natural language processing |
| RCT | randomized controlled trial |
| UK | United Kingdom |
| US | United States |
| WHO | World Health Organization |

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
