# Peer review of "AI Chatbots in Digital Mental Health"

_informatics, doi:10.3390/informatics10040082_

Round 1

Reviewer 1 Report

Comments and Suggestions for Authors

This is a fairly well written report on an important and interesting matter. The methodology used (an apparently selective literature review) is inadequate as it is highly suspect of bias. Hence the paper should be redone as a scoping or systematic review with a PRISMA diagram as part of that. 

Comments on the Quality of English Language

A few typo corrections needed. 

Author Response

Comment 1: This is a fairly well written report on an important and interesting matter. The methodology used (an apparently selective literature review) is inadequate as it is highly suspect of bias. Hence the paper should be redone as a scoping or systematic review with a PRISMA diagram as part of that.

Response 1: 

A narrative literature review was selected rather than a scoping or systematic review because the latter reviews were unsuccessful due to heterogeneity of search results. The method described for conducting the literature review, based on the four steps outlined by Demiris et al, was successful for several reasons. First, by searching various databases and search engines, I was able to access a wide range of relevant articles, ensuring a comprehensive exploration of the topic. This helped in gathering a diverse set of perspectives and insights. Second, identifying keywords from the relevant articles allowed me to focus the search and find articles specifically related to AI chatbots in digital mental health. This helped avoid irrelevant or unrelated studies, saving time and effort. Third, reviewing the abstracts and full texts of the identified articles allowed me to filter out articles that did not align with the research aim. This ensured that only the most relevant articles were included in your review, enhancing the quality and focus of the findings. Lastly, summarizing and synthesizing the findings from the selected articles helped me integrate the information in a coherent and organized manner. This approach allowed me to present a comprehensive overview of the existing literature on AI chatbots in digital mental health.

However, I made an edit to better explain how the heterogeneity of the topic prevented a systematic review: “The topic of AI chatbots in digital mental health is still evolving, and there was not enough studies that meet the strict criteria of a systematic review. Additionally, the field of digital mental health is interdisciplinary, incorporating aspects of psychology, technology, and healthcare. This leads to a wide range of research approaches, methodologies, and study designs, making it challenging to apply strict systematic review criteria. Instead, a purposively selected articles approach showed an educational approach on how AI chatbots may impact digital mental health. By purposively selecting articles that aligned with the research aim and reviewing them in a comprehensive manner, valuable insights were gathered and presented in a coherent narrative literature review. This approach allowed for flexibility in considering various perspectives and ideas within the topic, contributing to a more holistic understanding of the subject matter”.

Reviewer 2 Report

Comments and Suggestions for Authors

The review of the theoretical framework is quite scarce and must be improved. It is recommended to delve deeper into the scientific literature on the concept and impact of artificial intelligence chatbots. Likewise, it is advisable to provide more information about the use of chatbots in the area of mental health, since it is addressed in a very superficial way. It is also recommended to include previous narrative reviews in the topic addressed. In general terms, it is necessary to give greater content to the theoretical and introductory framework of the research.

The general research objective is clearly defined. Furthermore, a clear reference is made to the research questions and their connection with the methodological section, appropriately being the starting point for the development of the narrative literature review.

The starting point of the methodological framework is adequate, taking as reference four key steps to be carried out within the review of narrative literature. However, there are many methodological and procedural aspects that must be mentioned to give scientific and methodological rigor to the review carried out:

• The selection criteria must be mentioned and the choice of keywords justified, while including the results obtained based on the use of Boolean operators in the searches or not.

• The methodological process used to choose the documents to be analyzed must be described (initial number of documents reviewed and final number with which the literary review has been carried out).

• The inclusion and exclusion criteria of the documents must be included.

• Reference data of the analyzed records must be tabulated, such as authorship, publication date, database obtained, etc.

The results are reflected as a discussion section rather than a results section. The research questions are answered based on references to the analyzed texts and scientific literature. The results do not show empirical work, but rather a simple writing of key aspects of the documents analyzed. It is difficult to establish substantial conclusions and findings, from aspects that are seen as merely descriptive and have no empirical value.

The conclusions are written in a very general way, with greater specificity of the findings found being necessary, as in the mention: "The development of AI chatbots brings with it opportunities and challenges." It would be advisable to establish which opportunities and which most relevant challenges have been identified. found in the review, “the review highlights the power of collaborative AI to avoid bias” it would be advisable to establish how AI avoids bias. Furthermore, it is necessary to include the limitations and future lines of research derived from this work, since there is no mention of these sections in the conclusions.

Author Response

Thanks for the comments. Please see each comment then the response below.

Comment 1: The review of the theoretical framework is quite scarce and must be improved. It is recommended to delve deeper into the scientific literature on the concept and impact of artificial intelligence chatbots.

Response 1: The paper responds to predefined questions, and Section 3.1, ‘The impact of AI chatbots on technology integration’, describes both the concept and impact of AI chatbots on mental health. References 17-20 describe the concept for AI chatbots generally and references 21-31 are specific to mental health, which is used as an example. For a narrative literature review, this is sufficient to introduce the topic. Reference 18 shows a recent reference from an overview of AI mental health chatbots showing use in “accessible, affordable and convenient social and psychological support” although “crisis support” is insufficient because of “poor semantics”. This requires qualitative studies, which could use AI chatbots for NLP. Therefore, there is a clear concept for AI chatbots generally, more specifically for mental health in psychological support roles. However, there are limitations for crisis support at present which requires attention to the human factors of using AI chatbots. The concept and impact is clearly stated and direction for future research is provided.

Comment 2: Likewise, it is advisable to provide more information about the use of chatbots in the area of mental health, since it is addressed in a very superficial way.

Response 2: Section 3.1 outlines the following overview addressing the area of mental health: “A 2023 overview of mental health chatbots found 10 apps on the market for a variety of mental health concerns (e.g., anxiety and depression) and users (e.g., rural dwellers, shift workers, students, veterans and adolescents) for a variety of aims (e.g., to improve social or job interviewing skills) [18]”. Therefore, it shows that AI chatbots are used for variety of mental health issues, populations and aims (the conclusion was revised to include this summary). Reference 18 is up-to-date and sufficient to show the use of AI chatbots. There is no need to duplicate efforts. More detail in the proceeding paragraphs in Section 3.1 is provided about the limitations of human factors, the need to bring together disparate differences among the various professionals involved, and the promise of certain chatbots such as Wysa and Woebot. However, for crisis support there has been concerns e.g., Eliza. I then pointed to the need for qualitative studies to increase multidimensional feedback synthesis and to help reduce poor semantics and errors as well as increase trust in AI chatbots. This amounts to an indepth analysis of the mental health applications of AI chatbots.

The following was added to the conclusion: “The development of AI chatbots brings opportunities in serving underserved and unserved populations, as well as blending care for the well-served especially in treating common disorders such as anxiety and depression. However, there are challenges in providing the level of care require for crisis support so the human factors of human-computer interaction require more attention through empirical research”.

Comment 3: It is also recommended to include previous narrative reviews in the topic addressed. In general terms, it is necessary to give greater content to the theoretical and introductory framework of the research.

Response 3: There are no preceding narrative reviews on the mental health AI chatbots. Therefore, the current research introduces synthesizing the topic, selecting a mix of theoretical and empirical research to show the potential for following studies to provide an introductory framework. The following was added to the conclusion: “The novelty of AI chatbots in mental health means this narrative literature review shows examples of theoretical and empirical research that future studies can apply”.

Comment 4: The general research objective is clearly defined. Furthermore, a clear reference is made to the research questions and their connection with the methodological section, appropriately being the starting point for the development of the narrative literature review.

The starting point of the methodological framework is adequate, taking as reference four key steps to be carried out within the review of narrative literature. However, there are many methodological and procedural aspects that must be mentioned to give scientific and methodological rigor to the review carried out:

  • The selection criteria must be mentioned and the choice of keywords justified, while including the results obtained based on the use of Boolean operators in the searches or not.

Response 4: Please see the added selection criteria and use of Boolean operators in the Methods:

Comment 5:

  • The methodological process used to choose the documents to be analyzed must be described (initial number of documents reviewed and final number with which the literary review has been carried out).

Response 5: It is not a systematic or scoping review. Therefore, this is not required procedure for a narrative literature review. My previous papers with Informatics, e.g., The Impact of YouTube on Loneliness and Mental Health, did not use this procedure and that is reflected in other papers. I have deleted the (n=) as it was superfluous to this type of research.

Comment 6:

  • The inclusion and exclusion criteria of the documents must be included.

Response 6: Please see Methods – this has been included.

Comment 7:

  • Reference data of the analyzed records must be tabulated, such as authorship, publication date, database obtained, etc.

Response 7: This is not required for a narrative literature review. A systematic or scoping review is beyond the scope of this topic/research.

Comment 8:

The results are reflected as a discussion section rather than a results section. The research questions are answered based on references to the analyzed texts and scientific literature. The results do not show empirical work, but rather a simple writing of key aspects of the documents analyzed. It is difficult to establish substantial conclusions and findings, from aspects that are seen as merely descriptive and have no empirical value.

Response 8: This is because narrative literature review is a type of review that summarizes and synthesizes the existing literature on a topic, without using a systematic or scoping approach. Narrative reviews are often more subjective and less structured than systematic or scoping reviews, and it relied on my expertise and interpretation of the literature. Narrative reviews are useful for providing an overview of a topic, identifying gaps in the literature, and generating new research questions. As such I added the following to the conclusion: “This review is useful for providing an overview of the topic, identifying gaps in the literature, and generating new research questions”.

In contrast, systematic and scoping reviews are more structured and objective approaches to reviewing the literature. Systematic reviews use a rigorous and transparent methodology to identify, appraise, and synthesize all relevant studies on a specific research question. Scoping reviews are broader in scope than systematic reviews and aim to map the existing literature on a topic, rather than answer a specific research question.

I detailed a mix of theoretical and empirical research on the use of AI chatbots in mental health. The study reviewed existing literature on the topic, including empirical studies that evaluated the effectiveness of AI chatbots in mental health care. The study also discussed theoretical frameworks that can be used to guide the development and evaluation of AI chatbots in mental health care. I added the following to the conclusion: “By synthesizing both theoretical and empirical research, the study provided a comprehensive overview of the current state of AI chatbots in mental health care”.

While some studies reviewed were conceptual in nature, others were empirical studies that evaluated the effectiveness of AI chatbots in mental health care. For example, see references 22-27 and 18.

I added the following to the conclusion: “Overall, the study shows promise for the use of AI chatbots in mental health care, but also highlights the need for further research to evaluate their effectiveness and address ethical concerns”.

Comment 9:

The conclusions are written in a very general way, with greater specificity of the findings found being necessary, as in the mention: "The development of AI chatbots brings with it opportunities and challenges." It would be advisable to establish which opportunities and which most relevant challenges have been identified.

Response 9: The conclusions have been revised (see mark up). E.g. “The development of AI chatbots brings opportunities in serving underserved and unserved populations, as well as blending care for the well-served especially in treating common disorders such as anxiety and depression. However, there are challenges in knowing which AI chatbots are of good quality, useful and effective. Therefore, it is important for future research to clarify on these areas as well as the level of care re-quired for crisis support. The human factors of human-computer interaction require more attention through empirical research”.

Comment 10:

found in the review, “the review highlights the power of collaborative AI to avoid bias” it would be advisable to establish how AI avoids bias.

Response 10: I have included the following additional text to demonstrate tackling bias. “Overall, the method shows a systematic and transparent approach that minimizes bias by ensuring a comprehensive search, focusing on relevant articles, and presenting a fair synthesis of findings. However, it’s important to note that bias can still exist in the literature itself, so the studies were critically evaluated and any potential limitations or biases within the selected articles were acknowledged”.

Comment 11: Furthermore, it is necessary to include the limitations and future lines of research derived from this work, since there is no mention of these sections in the conclusions.

Response 11: This has been partly addressed in the existing text: “Nevertheless, limitations such as poor semantics, biases, and the need for qualitative studies to improve user experience must be acknowledged and addressed. AI chatbots should be seen as complementary tools rather than replacements for human mental health professionals”. I added the following: “However, there is a need for more empirical evidence and advocacy for users and practitioners to distinguish the quality, usability and effectiveness and for what uses and populations. If AI chatbots evolve to provide appropriate answers to these areas for clarification, then more autonomous use of these tools will become progressively possible”.

Reviewer 3 Report

Comments and Suggestions for Authors

This is a very interesting paper, appropriately structured and documented. Research questions are appropriately expressed and corresponding literature-based answers well formulated. Opposite views between technology positions and orientations and in the field (digital mental health), observations are formulated with precision. The field positions are privileged and I agree with this orientation.

In conclusion statement “AI chatbots should be seen as complementary tools rather than replacements for human mental health professionals.” is at least today appropriate position, which could evolve in the future if indicated evolutions receive appropriate answers. More autonomous working of tools will become progressively possible.   

Concreate examples and actions are mainly based on Australian context. It could be appropriate to enlarge them to more worldwide scope as, for example, European Commission contributions by European Digital Services Act, with European Regulators’ Group for Audiovisual Media Services – ERGA

https://digital-strategy.ec.europa.eu/en/policies/digital-services-act-package

Author Response

Response 1: Thank you for the understanding that AI chatbots hold promise for mental health, however, there is a need for clarification about uses in terms of quality, usability and effectiveness. I added the following to the conclusion: “If AI chatbots evolve to provide appropriate answers to these areas for clarification, then more autonomous use of these tools will become progressively possible”.   I added the reference from European Commission: “The EU implemented the Digital Services Act and the Digital Market Act aiming to create a safer digital space where the fundamental rights of users are protected and to establish a level playing field for businesses [59]”.

Round 2

Reviewer 1 Report

Comments and Suggestions for Authors

My concern about the methodology involving a selective rather than a scoping literature review and hence risking considerable bias has not been adequately addressed in the revision. 

Comments on the Quality of English Language

I have no major concern with the language. 

Author Response

Comment 1: My concern about the methodology involving a selective rather than a scoping literature review and hence risking considerable bias has not been adequately addressed in the revision.

Response 1: I appreciate the suggestion for a scoping review and gave this careful consideration. My previous paper, “Evaluation of the Use of Digital Mental Health Platforms and Interventions: Scoping Review” shows how this is a preferable review including tabulation of results when there is the quantity and quality of empirical evidence available. I determined that it is not yet possible to produce a scoping review for AI chatbots in mental health because it is simply not advanced enough yet in the literature. Hence, the narrative literature review should be seen as a “stepping stone” to a scoping review.

In regard to the concern over a “selective rather than a scoping literature review”, narrative reviews are useful for providing an overview of a topic, identifying gaps in the literature, and generating new research questions. As such I added the following to the conclusion: “This review is useful for providing an overview of the topic, identifying gaps in the literature, and generating new research questions”.

In contrast, scoping reviews broadly aim to map the existing literature on a topic, rather than answer a specific research question.

I detailed a mix of theoretical and empirical research on the use of AI chatbots in mental health. The study reviewed existing literature on the topic, including empirical studies that evaluated the effectiveness of AI chatbots in mental health care. The current study also discussed theoretical frameworks that can be used to guide the development and evaluation of AI chatbots in mental health care. I added the following to the conclusion: “By synthesizing both theoretical and empirical research, the study provided a comprehensive overview of the current state of AI chatbots in mental health care”.

While some studies reviewed were conceptual in nature, others were empirical studies that evaluated the effectiveness of AI chatbots in mental health care. For example, see references 22-27 and 18.

I added the following to the conclusion: “Overall, the study shows promise for the use of AI chatbots in mental health care, but also highlights the need for further research such as scoping reviews to evaluate their effectiveness and address the risk of bias and ethical concerns”.

In regard to the concern about the risk of bias: While selective literature reviews may be more prone to bias than systematic reviews, they are still valid forms of research. For example, the methodology used in my narrative literature review was appropriate for answering the research question and provided a comprehensive exploration of the topic. However, I addressed this concern further, adding the following to the Methods section: “Overall, the method shows a systematic and transparent approach that minimizes bias by ensuring a comprehensive search, focusing on relevant articles, and presenting a fair synthesis of findings. However, it’s important to note that bias can still exist in the literature itself, so the studies were critically evaluated and any potential limitations or biases within the selected articles were acknowledged”.

I also made a change in the abstract, making a call for scoping reviews: “The integration of HAI principles, responsible regulation, and scoping reviews are crucial to maximizing their benefits while minimizing potential risks”. This places the current study as a stepping stone to scoping review research which may be possible in the next years (if and when there is less subject heterogeneity and novelty).

Reviewer 2 Report

Comments and Suggestions for Authors

The aspects required in the previous review have been improved and justified in a coherent and reasoned manner based on the thematic and methodological field.

Author Response

Thank you for supporting the improvements and justifications made in the revision, and for noting it is practical and reasonable with regards to the thematic and methodological field.